# Using machine learning to investigate the relationship between domains of functioning and functional mobility in older adults

**Keisuke Hirata** 🄐 *****, Makoto Suzuki, Naoki Iso, Takuhiro Okabe, Hiroshi Goto, Kilchoon Cho, Junichi Shimizu**

Faculty of Health Sciences, Tokyo Kasei University, Saitama, Japan

***** hirata-ke@tokyo-kasei.ac.jp

## Abstract

Previous studies have shown that functional mobility, along with other physical functions, decreases with advanced age. However, it is still unclear which domains of functioning (body structures, body functions, and activities) are most closely related to functional mobility. This study used machine learning classification to predict the rankings of Timed Up and Go tests based on the results of four assessments (soft lean mass, $FEV_1/FVC$, knee extension torque, and one-leg standing time). We tested whether assessment results for each level could predict functional mobility assessments in older adults. Using support vector machines for machine learning classification, we verified that the four assessments of each level could classify functional mobility. Knee extension torque (from the body function domain) was the most closely related assessment. Naturally, the classification accuracy rate increased with a larger number of assessments as explanatory variables. However, knee extension torque remained the highest of all assessments. This extended to all combinations (of 2–3 assessments) that included knee extension torque. This suggests that resistance training may help protect individuals suffering from age-related declines in functional mobility.

## 1. Introduction

Fractures account for approximately 10% of all cases in which older individuals become bedridden [1]. As Japan is the most rapidly aging society in the world, it is a matter of social concern to help prevent falls among older adults. One solution is to predict the probability of functional mobility decline by assessing physical function [2]. Moreover, declines in functional mobility and balance are related to falls [3, 4]. Therefore, previous studies have identified the predictors of results for the Timed Up and Go (TUG) test [2], maximum walking speed test [5], one-leg standing time test [6], functional reach test [7], Berg balance scale [8], functional balance scale [9], and four square step test [10]. Among these, the TUG test is highly recommended as a screening tool for identifying whether older individuals are at risk of falling [11]. The TUG test is also useful for predicting functional mobility, especially balancing and gait maneuvers used in everyday life (e.g., standing up, sitting down, walking, and turning) [2].

**Data Availability Statement:** The data underlying the results presented in the study are available from Figshare (DOI: 10.6084/m9.figshare.13553903.v1).

**Funding:** The author(s) received no specific funding for this work.

**Competing interests:** The authors have declared that no competing interests exist.

A clinical review by Brown et al. [12] found that the predictors of mobility limitation could be aggregated into five factors: age, physical activity, BMI, muscle strength, and disease. Therefore, in older adults, mobility has a complex relationship with various domains of functioning. Further, previous studies have found that poor TUG ranks are associated with the affected domain of body structures (e.g., body composition [13]), body functions (e.g., respiratory function [14], muscle strength [15]), and activities (e.g., balance ability [16]). Thus, relevant literature shows that there are correlations between each domain of functioning (body structures, body functions, and activities) in healthy older adults [17–19]. The domains of functioning were defined in the International Classification of Function, Disability, and Health (ICF) by the World Health Organization (WHO) in 2015. However, regarding older adults in Japan, it remains unclear which functioning assessments are most closely related to functional mobility. Additionally, it is difficult to interpret functioning assessments of certain older individuals due to the increased frequency of multiple chronic diseases [20] and general declines in overall functioning [21, 22]. Moreover, frailty consists of several interrelated factors, including age-associated declines in lean body mass, strength, endurance, balance, and walking performance [23]. In sum, it remains unclear which domain of functioning is most related to TUG rank, which makes it difficult to determine the priority of results when attempting to interpret multiple assessments for a given individual.

To bridge the aforementioned gaps in the literature, we used support vector machines (SVMs) of machine learning classification algorithm (commonly used for estimating multivariate patterns) [24]. SVMs are suitable for finding relationships by high-dimensional mapping using support vectors from small sample data with complex relationships. Moreover, the prediction accuracy of machine learning prediction methods (e.g., Random Forest, Artificial Neural Network), including SVMs, are not affected by multicollinearity [25]. When dealing with multivariates that are correlated with each other such as in the present study, one of the reasonable methods to employ is SVMs [26]. The purpose of our study was clarified assessment patterns for each domain reflecting functional mobility rankings according to the TUG test. As such, we employed a machine learning classification by using TUG rank as an objective variable, while the assessment values of each domain (body structures, body functions, and activities) relative to the averages of the same generations were set as explanatory variables. From the viewpoint of structure of ICF, our hypothesis is that the prediction accuracy increases in the order of the domain from the bottom to the top, and the activities similar to TUG is the highest. Based on this classification, we attempted to clarify which assessments were most closely related to functional mobility among older adults. To the best of our knowledge, no previous studies have investigated this issue in Japan. Our study on Japanese older adults is relevant because Japan has one of the highest life expectancy rates in the world.

## 2. Materials and methods

### 2.1 Eligibility criteria

The East Japan Community Study of Aging (EJCSA) is an ongoing longitudinal survey that targets home-dwelling, healthy older adults in eastern Japan. The number of data recorded in the database of the EJCSA from 2018 to 2020 was 121. A total of 112 subjects were extracted as the target data for analysis according to the criteria of aged over 50 years and without physical disabilities.

The sample size was determined following a previous study finding knee extension torque/bodyweight of $1.91 \pm 0.58$ Nm/kg (average ± standard deviations) among 24 middle age and $1.55 \pm 0.47$ Nm/kg among 24 old adults [27]. Sample size was calculated based on a desired 80% statistical power to detect a 0.35 Nm/kg difference (standard effect size, 0.60) in knee

extension torque. We confirmed that the sample size was satisfied after being estimated using G*Power 3.1 software (Franz Faul, University of Kiel, Kiel, Germany), with an effect size 0.30, a minimum power 0.80, and $\alpha = 0.05$. All participants provided written informed consent prior to participation. Furthermore, all procedures adhered to the Declaration of Helsinki. The experimental procedures were specifically approved by the Research Ethics Committee of Tokyo Kasei University.

## 2.2 Classifications: Five groups and four assessments

Participants performed the TUG test at a comfortable speed. We then used their results to classify them into five groups, following the guidelines that the Tokyo Metropolitan Institute of Gerontology established based on a study by Obuchi et al. [28]. These were labeled Group 1 (males $\geq$ 7.2; females $\geq$ 8.9), Group 2 (males 6.1–7.1; females 7.5–8.8), Group 3 (males 5.5–6.0; females 6.5–7.4), Group 4 (males 5.0–5.4; females 5.8–6.4), and Group 5 (males $\leq$ 4.9; females $\leq$ 5.7). All units indicate seconds.

The assessments were used as follows.

1. *Soft lean mass (SLM)*
   Body composition was assessed through direct segmental multi-frequency bioimpedance analysis (DSM-BIA) using the InBody770 (InBody Co., Ltd., Korea), which uses a multi-frequency segmental measurement method with an eight-point tactile electrode. Multi-frequency measurements were taken using frequencies of 1, 5, 50, 250, 500, and 1000 kHz for each body segment. The data were normalized according to generation and sex, based on Lee's work [29].

2. *$FEV_1$/FVC of respiratory function*
   $FEV_1$/FVC ratios were measured using a digital spirometer (AS-407, MINATO Medical Science Co., Ltd., Japan). Participants were asked to take deep breaths using a mouthpiece attached to the spirometer, while sitting. They were then asked to hold their breath long enough to seal their lips tightly around the mouthpiece. Afterwards, they were asked to hold their noses tightly and exhale the air out as forcibly and quickly as possible, until all the air was expelled. Participants were verbally encouraged to continue exhaling during this phase.

3. *Knee extension torque (KET)*
   Participants were tested for isometric maximal voluntary contraction of the knee extensor muscles on the dominant lower limb using a dynamometer (μ Tas-01, Anima Co., Ltd., Japan). During the test, participants were seated comfortably in a chair with their trunks erect, while knee and hip angles were maintained at 90 degrees. The task consisted of a quick increase to the maximum force exerted by the leg. The higher data of each leg was divided by the participants' weight and normalized according to generation and sex, based on Bohannon's work [30].

4. *One-leg standing time (OLST)*
   The OLST test measures participants' ability to continue standing on one leg with their eyes closed. The test ended when participants moved their supporting legs, lost balance, or opened their eyes. The upper threshold was 30 seconds. We registered the mean times for each supporting leg and normalized them according to generation and sex, based on Springer's work [31].

The $FEV_1$/FVC, KET, and OLST were adopted higher result in two trials. The participants had enough rest time between each trial, which involved resting for at least about 3 minutes

between each trial, taking into account the effects of muscle fatigue [32]. The participants completed the assessments randomly. All assessments were normalized by each maximum value and expressed as 0–1.

## 2.3 Data analysis

We performed a one-way ANOVA for age differences among the TUG groups. Correlation coefficient was estimated the relationship between data of each assessment before normalizing and result of TUG.

All five TUG groups were used as objective variables in the SVM. To avoid group variability, data from each group included four assessments: (1) SLM, (2) $FEV_1$/FVC, (3) KET, and (4) OLST. The data of each group were randomly expanded to 1,000 data to overcome the limitation of a small sample size and difference in sample sizes among groups. Therefore, a total of 5,000 data measurements from all groups were maintained for the distribution of actual data using the bootstrap method. This bootstrap resampling method is often used in demographic studies [33].

The data were randomly divided as 90% training data and 10% testing data. The training data were submitted to the SVM; that is, the SVM algorithm was constructed as a prediction model using the training data. After the training, the resultant SVM prediction model with the 90% training data (4500 data) was used to predictively classify the remaining 10% testing data (500 data) into either of the five TUG groups or the four assessments for cross-validation. To verify the effect of the number and/or combination of assessments on predictive accuracy, SVM prediction was performed for all 15 combinations of the four assessments ($_4C_1 + {}_4C_2 + {}_4C_3 + {}_4C_4$), and randomly divided into training and testing data for each prediction.

Predictive accuracy was calculated as the total number of successful predictions in each group divided by the total number of predictions in all groups. This ensured that a trained SVM with a Gaussian kernel could prospectively be generalized ($G(x_j, x_k) = \exp(-\|x_j - x_k\|^2)$). In the current investigation, we used "templateSVM," available in MATLAB software (The MathWorks Inc., Natick, MA, USA), which utilizes the algorithm defined by Schölkopf et al. [34]. The relationship between the accuracy rate and number of assessments was tested using AIC (Akaike's information criteria) as a non-linear regression equation.

## 3. Results

Table 1 shows participants' characteristics for each group. As seen, participants' ages in Group 1 were significantly higher than for participants in Groups 2, 3, and 4 ($p < 0.05$). There were significant differences between Groups 1 and 4 regarding SLM, KET, and OLST. Except for the SLM measurements between Groups 1 and 3 ($p < 0.05$), there were no significant intergroup differences for the other assessments. S1 Fig shows the relationships between the TUG test and other assessments (only KET showed a low negative correlation coefficient; $p < 0.05$), while Table 2 shows the average accuracy rates of SVM prediction. Combinations of three to four assessments had high rates. The relationship between the accuracy rate and number of assessments was highly correlated (S2 Fig, R = 0.89, $p < 0.001$). Notably, all top accuracy rates for single assessment and each combination of two and three assessments included KET (SLM + $FEV_1$ / FVC + KET 89.2%, and SLM + KET 82.2%, only KET 57.2%). Moreover, the combination of $FEV_1$ / FVC and KET was higher accuracy rate than the combination of SLM, $FEV_1$ / FVC and OLST + KET (82.0%).

**Table 1. Participants' characteristics and bootstrap resampling data for each TUG group (mean ± standard deviation).**

| | Group 1 | | Group 2 | | Group 3 | | Group 4 | | Group 5 | |
|---|---|---|---|---|---|---|---|---|---|---|
| | Male TUG ≧ 7.2 | | 6.1 ≧ Male TUG > 7.1 | | 5.5 ≧ Male TUG > 6.0 | | 5.0 ≧ Male TUG > 5.4 | | 4.9 ≧ Male TUG | |
| | Female TUG ≧ 8.9 | | 7.5 ≧ Female TUG > 8.8 | | 6.5 ≧ Female TUG > 7.4 | | 5.8 ≧ Female TUG > 6.4 | | 5.7 ≧ Female TUG | |
| Participants (n = 112) | 30 | | 32 | | 28 | | 18 | | 4 | |
| Age (years) | 74.7 ± 7.4 [*2, 3, 4] | | 68.8 ± 6.1 [*1] | | 65.3 ± 7.5 [*1] | | 66.9 ± 8.0 [*1] | | 69.8 ± 8.0 | |
| Sex (male / Female) | 25 / 5 | | 13 / 19 | | 2/26/2021 | | 0 / 18 | | 1 / 3 | |
| | Actual | Bootstrap | Actual | Bootstrap | Actual | Bootstrap | Actual | Bootstrap | Actual | Bootstrap |
| 1) Soft lean mass (kg) | 24.05 ± 3.93 [*3, 4] | 24.07 ± 3.85 | 21.86 ± 5.15 | 21.89 ± 5.03 | 20.26 ± 2.92 [*1] | 20.24 ± 2.83 | 19.68 ± 1.56 [*1] | 19.67 ± 1.50 | 21.78 ± 4.94 | 21.81 ± 4.03 |
| 2) $FEV_1$ / FVC (%) | 69.70 ± 11.86 | 69.63 ± 11.59 | 75.18 ± 9.99 | 75.19 ± 9.75 | 74.57 ± 8.64 | 74.59 ± 8.39 | 76.28 ± 9.77 | 76.25 ± 9.26 | 74.75 ± 4.65 | 74.76 ± 3.78 |
| 3) Knee extension Torque (Nm / kg) | 0.91 ± 0.31 [*4] | 0.92 ± 0.30 | 0.99 ± 0.26 | 0.99 ± 0.25 | 1.13 ± 0.30 | 1.12 ± 0.29 | 1.24 ± 0.22 [*1] | 1.24 ± 0.22 | 1.07 ± 0.36 | 1.07 ± 0.30 |
| 4) One leg standing time (sec) | 19.75 ± 9.34 [*3] | 19.86 ± 9.14 | 22.41 ± 10.06 | 22.41 ± 9.85 | 26.53 ± 5.04 [*1] | 26.58 ± 4.91 | 26.10 ± 7.32 | 26.12 ± 6.95 | 28.25 ± 3.50 | 28.24 ± 2.53 |

[*] $p < 0.05$; n = 112 older Japanese adults.

The number of next to [*] is the group number with a significant difference.

## 4. Discussion

We conducted SVM prediction for TUG ranks based on all 15 combinations of four assessments in each domain (body structures: soft lean mass; body functions: $FEV_1$ / FVC, knee extension torque; activities: one-leg standing time). KET still had the highest accuracy rate in any single assessment, and all combinations (of 2–3 assessments) that included KET were the highest. To the best of our knowledge, this is the first study to identify the most closely related functioning assessment to functional mobility using a machine learning classification method.

The accuracy rate increased depending on the number of explanatory variables as assessments (S2 Fig). This result is natural because all assessments are known to contribute to the TUG. Next, in combinations of three assessments, the OLST, $FEV_1$ / FVC and SLM assessments were omitted in order. The accuracy rate of the combination of SLM, $FEV_1$ / FVC, and

**Table 2. List of average accuracy rates (highest to lowest).**

| 1) Soft lean mass | 2) $FEV_1$ / FVC | 3) Knee extension torque | 4) One leg standing time | Accuracy rate (%) |
|---|---|---|---|---|
| ○ | ○ | ○ | ○ | 94.4% |
| ○ | ○ | ○ | - | 89.2% |
| ○ | - | ○ | ○ | 88.4% |
| - | ○ | ○ | ○ | 87.8% |
| ○ | - | ○ | - | 82.2% |
| ○ | ○ | - | ○ | 82.0% |
| - | - | ○ | ○ | 79.4% |
| ○ | - | - | ○ | 75.0% |
| - | ○ | ○ | - | 73.6% |
| - | ○ | - | ○ | 66.8% |
| ○ | ○ | - | - | 64.8% |
| - | - | ○ | - | 57.2% |
| ○ | - | - | - | 56.6% |
| - | - | - | ○ | 50.8% |
| - | ○ | - | - | 46.2% |

Note: Open circles indicate adopted assessments, while minuses indicate non-adopted assessments.

OLST were lower than that of the combination of two assessments including KET. However, this accuracy rate did not depend on the number of assessments. Moreover, the bottom three in the single assessments were SLM, $FEV_1$ / FVC, and OLST. From these points, the difference in the explanatory variables can be seen between SLM, $FEV_1$ / FVC, and OLST compared to KET. However, since the chance of predicting the TUG's five ranks by simply thinking is 20%, it cannot be said that the predictive accuracy of these evaluations is low (SLM: 56.6%; $FEV_1$ / FVC: 50.8%; KET: 46.2%). This may be due to the fact that the evaluation is related with TUG.

## 4.1 Significance as an analytical method for the support vector machine

Among four assessments, our results showed that knee extension torque was the most closely related to each participant's TUG rank. The various abovementioned parameters of functioning typically decrease with age [20]. Personal data obtained from older adults often include high-correlation (multicollinearity) data, such as age-dependent parameters [23]. Using a multivariate analysis method precludes the analysis of data unless the other explanatory variables are removed. However, machine learning prediction can make it possible to analyze data with multicollinearity [35]. One of the methods of machine learning prediction is SVMs with kernel function. Therefore, results achieved through SVM projection are not affected by collinearity [36]. In other words, this machine learning prediction method enabled us to avoid removing important information about participants. Moreover, the input data are first projected onto a higher dimensional space before they are employed in the estimation process. Thus, this method allowed us to conduct a complex factor analysis by multiple variables. Other studies have used SVMs to analyze the relationships between outcomes and multiple complicating factors, as to individually predict each participant's prognosis [37]. An SVM with high discrimination accuracy was suitable for this study due to its usefulness in selecting evaluations with complicated correlations. However, the classification function obtained through the SVM is a black box that outputs only the classification results. This makes it difficult to interpret the contributions of each variable. In this study, all explanatory variables (SLM, KET, $FEV_1$/FVC, and OLST) were already known contributors to the objective variable (TUG), as mentioned above. For this reason, the study was able to use machine learning to demonstrate a relationship between knee extension muscle strength as a body function domain and the TUG test, which analyses of variance and correlations cannot reveal.

## 4.2 The most related domain of functioning assessment is body function: Knee extension torque

In our study, KET as the evaluation of knee extension muscle strength was the assessment most related with functional mobility. It should be noted that our results differed from those of previous studies. In older adults with functional limitations, previous studies have found that neither muscle strength nor power in the lower extremities were correlated with walking distance [38]. In older adults with high activity, another study reported that leg muscle strength and leg lean tissue mass are not outcomes for predicting mobility, because both are similarly weakly correlated with gait performance [39]. The results of the aforementioned studies likely differed from ours due to differences in methodology (short-term interventions and correlation coefficients analysis vs. machine learning).

Changes to the neural system and muscle fibers, which naturally occur with age, lead to declined neuromuscular function [40]. This is associated with a reduced ability to generate both muscle strength and power, consequently impeding daily living activities [41, 42]. In our study, body structure comparisons between SLM and $FEV_1$/FVC showed that SLM was significantly higher in Group 1 (low rank), while there were no significant intergroup differences for

$FEV_1/FVC$. Compared to cardiorespiratory fitness, knee muscle torque is significantly associated with overall physical activity, postural transitioning, walking, and stair climbing [27]. Similarly, it is conceivable that the relationship between gait performance and leg muscle strength is stronger than that between gait performance and leg muscle mass [39]. The European Working Group on Sarcopenia in Older People suggests a conceptual staging that includes presarcopenia, sarcopenia, and severe sarcopenia. The presarcopenia stage is characterized by decreased muscle mass without significant effects to muscle strength or physical performance. This stage can only be identified using techniques that accurately measure muscle mass and reference standard populations [43]. As it is possible that many of this study's participants also had presarcopenia, they may have had a reduction in SLM, but not reductions in TUG ranks or knee muscle strength.

The functional reach test is another common activity assessment when evaluating dynamic balance. However, a previous study found that knee extension muscle strength was a more important independent factor than functional reach [44]. Therefore, we conducted comparisons between activities via the one-leg standing test. Since the TUG test reflects walking speed, it may also be associated with activity tasks, such as the 10 m walking test. In a previous study, approximately 50% of older adults who had no difficulties when turning achieved nearly the same results as younger adults with similar characteristics. In said study, researchers used a pivot strategy involving one or two steps to accomplish turns in 2.49 seconds or less with no signs of imbalance [45]. For both young and old participants without walking difficulty, the ratio of walking time to total time in the TUG test was nearly the same. This indicates that both the TUG test and KET are related to walking speed, suggesting a strong relationship between the TUG test and KET.

## 5. Conclusions and limitations

This study had some limitations. As previously mentioned, we could not include the causality between each domain/assessment and functional mobility through this study's machine learning method. Specifically, it focused only on four assessments throughout the three domains of functioning. Therefore, SVM prediction could not exclude the possible influences of other domains (e.g., environmental and individual factors) or assessments (e.g., cardiopulmonary function or outdoor activity) and/or any interactions between these factors. In addition, there are various machine learning classification methods such as random forest and artificial neural network available today. It is necessary to verify the optimal method for such assessments in future studies.

Although this study assessed limited domains of functioning, it strongly suggests that evaluations of body function are helpful when implementing preventive rehabilitations aimed at functional mobility. As resistance training can be used to maintain muscle strength, our results also suggest that it can help prevent age-related decline in functional mobility, thereby reducing the fall risk. If physiotherapists investigate the possibility of falling among highly active elderly people, prioritizing the monitoring of items related to physical function, especially muscle weakness could be valid. In making a rehabilitation program to reduce fall risk among the elderly, increasing the proportion of strengthening or maintaining muscle strength could be valid as part of rehabilitation therapy.

As the Japanese population continues to age, it is even more important to ensure that citizens maintain knee extension muscle strength at above-average levels for their respective age groups. This may help prevent the risk of older adults becoming bedridden, while also reducing nursing care requirements and lowering overall medical expenses. For these reasons, the

same measures are also important in other countries, and require further investigation, especially in countries with aging populations.

## Supporting information

**S1 Fig. Relationships between TUG times and assessments.** (A) soft lean mass; (B) knee extension torque; (C) $FEV_1$ / FVC; (D) one-leg standing time. Images show correlation coefficients (r) and linear regression lines.
(TIFF)

**S2 Fig. Relationships between accuracy rate and number of assessments.** The circles show the accuracy rate of each combination of assessments and the dashed line represents the non-linear regression line.
(TIFF)

## Acknowledgments

We thank Yoshiko Shibata for assisting with the data collection process.

## Author Contributions

**Conceptualization:** Keisuke Hirata, Makoto Suzuki, Naoki Iso, Takuhiro Okabe, Hiroshi Goto, Kilchoon Cho, Junichi Shimizu.

**Data curation:** Keisuke Hirata, Naoki Iso, Hiroshi Goto, Kilchoon Cho, Junichi Shimizu.

**Formal analysis:** Keisuke Hirata, Makoto Suzuki.

**Funding acquisition:** Keisuke Hirata, Takuhiro Okabe, Junichi Shimizu.

**Investigation:** Keisuke Hirata, Naoki Iso, Takuhiro Okabe, Hiroshi Goto, Kilchoon Cho, Junichi Shimizu.

**Methodology:** Keisuke Hirata, Makoto Suzuki.

**Project administration:** Keisuke Hirata, Takuhiro Okabe, Junichi Shimizu.

**Resources:** Keisuke Hirata.

**Software:** Keisuke Hirata.

**Supervision:** Keisuke Hirata.

**Validation:** Keisuke Hirata.

**Visualization:** Keisuke Hirata.

**Writing – original draft:** Keisuke Hirata.

**Writing – review & editing:** Keisuke Hirata, Makoto Suzuki, Naoki Iso, Takuhiro Okabe, Hiroshi Goto, Kilchoon Cho, Junichi Shimizu.

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
