## [Decision Letter · Decision Letter 0]

24 Dec 2020

PONE-D-20-33162

Using machine learning to investigate the relationship between domains of functioning and functional mobility in older adults

PLOS ONE

Dear Dr. Hirata,

Thank you for submitting your manuscript to PLOS ONE. After careful consideration, we feel that it has merit but does not fully meet PLOS ONE’s publication criteria as it currently stands. Therefore, we invite you to submit a revised version of the manuscript that addresses the points raised during the review process.

Three experts in the field have carefully evaluated the manuscript entitled, “Using machine learning to investigate the relationship between domains of functioning and functional mobility in older adults”. Their comments are appended below.

They gave positive comments for publication with leaving several concerns which should be considered before publication.

This Academic Editor believes that your revision will be sure to strengthen your manuscript. I am expecting to receive your replies to each critiques and necessary revision.

We look forward to receiving your revised manuscript.

Kind regards,

Manabu Sakakibara, Ph.D.

Academic Editor

PLOS ONE

Journal Requirements:

2. Please include a separate caption for each figure in your manuscript.

3. Please include your tables as part of your main manuscript and remove the individual files. Please note that supplementary tables should be uploaded as separate "supporting information" files.

Reviewers' comments:

Reviewer's Responses to Questions

**Comments to the Author**

1. Is the manuscript technically sound, and do the data support the conclusions?

Reviewer #1: Yes

Reviewer #2: Yes

Reviewer #3: Yes

2. Has the statistical analysis been performed appropriately and rigorously? 

Reviewer #1: No

Reviewer #2: Yes

Reviewer #3: Yes

3. Have the authors made all data underlying the findings in their manuscript fully available?

Reviewer #1: Yes

Reviewer #2: Yes

Reviewer #3: Yes

4. Is the manuscript presented in an intelligible fashion and written in standard English?

Reviewer #1: Yes

Reviewer #2: Yes

Reviewer #3: Yes

5. Review Comments to the Author

Reviewer #1: This study included 112 participants to conduct Support Vector Machine (SVM) prediction for five Timed Up and Go (TUG) rank-based groups, with the input of all 15 kinds of combinations of four assessments in each domain (body structures: soft lean mass; body functions: FEV1 / FVC, knee extension torque; activities: one-leg standing time). The results in the present study showed that knee extension torque was the most closely related to each participant’s TUG rank, and made the biggest contributions to an accurate prediction.

The study is of interest and the manuscript is written rather well. However, I have several comments, possibly helping to improve the overall scientific quality of the paper:

1, The authors should improve the introduction of their study with clear aims and hypotheses. Additionally, comparing with previous studies and clarifying the reasons for method selection, what are the benefits and drawbacks of SVM? The explanation of SVM on page 4, lines 65-66 is incorrect. The kernel function is not the way to avoid overfitting.

2, The machine learning method SVM was used in this study. However, functional mobility prediction is a research area with plenty of effective methods and algorithms that can be applied to obtain a highly accurate classification. Therefore, whether other much transparent methods (for example, the methods Random Forest (RF), Artificial Neural Network (ANN) or Partial Least Square Discriminant Analysis (PLS-DA)) may achieve better prediction and interpretable results than the SVM method but not used in this study need to be carefully discussed.

3, Moreover, a clear procedure of study should be claimed in the method part.

a) how many participants are exactly involved in as the input in the model? Because in the method section, the sample size was shown as 90, but in the results section, the input sample size was 121 or 112 (the statement was confusing);

b) it was not clear how and why the sample size needs to be extended to 1000 and what the benefits are; since the present study only included a maximum of four variables.

c) how the hyper-parameters of SVM were trained, which kernel function was used in the present study? Commonly, a valid hyper-parameter can be identified by using a validation set that is split from the 90% training set, or using a cross-validation method

d) whether all data as inputs to SVM model were standardized to make sure all variables have the same unit? Since multiple-unit variables can negatively impact the SVM model.

4, The output parameters of the prediction models in Table 2 only show accuracy. However, in order to show the model is valid and not overfitting, a comprehensive evaluation with other validation matrices should be provided, such as sensitivity, specificity and AUC.

5, On page 8, line 173, the statement is incorrect, SVM is not a kernel function, it was a classification or prediction method includes a kernel function. How the SVM addresses the multicollinearity of data in the present study (e.g., linear separately in high dimensional space) should be interpreted.

6, On page 8, lines 176-177, the expression “SVM projection enabled us to use information that was removed when selecting variables” is unclear. Who removes which variables? Or you mean no need for a prior variables’ selection?

7, On page 9, line 190, the statement “a larger number of explanatory variables” is not proper because you only show four variables in the present study.

8, On page 3, line 39, repeated references; lines 49-50, e.x. should be, e.g.,

The comments also attached as a PDF, please kindly refer to the attachment.

Reviewer #2: The authors implemented machine learning classification approach to predict the rankings of Timed Up and Go tests based on four assessments. The results showed that knee extension torque was the most closely related assessment. They concluded that the resistance training may prevent suffering from age-related declines in functional mobility.

1. Line 77. Please provide information how these 90 subjects were sampled from the original study?

2. Line 137. 10% were used as a testing data. Considering small sample size, isn’t 10% of sample is too small for testing?

3. Any evaluation on the results with the testing data?

4. Line 151. “A total of 121 participants were included” which is confusing when line 77 says including 90 healthy subjects. Please clarify this.

Reviewer #3: Overall I think the authors do a good job with this study. Here are my specific comments.

Introduction

Overall you did a good job with the introduction. I think you cover most of the relevant literature.

Methodology

Overall the methodology was well written, but I have some comments to allow for replicability

1. Was this retrospective data that you used a SVM model for? Or did you go recruit participants from this longitudinal study to collect the data for this specific study?

2. If these are participants who you recruited from the longitudinal study how did you recruit them? How did you contact them? What was the eligibility criteria to be in this particular study?

However, if these participants were part of the data collected for this longitudinal study, how did you select the participants?

3. When was this data collected?

4. What was the procedure? What was the order in which the participants completed the assessments?

5. You state that the participants had enough rest between trials, what is considered enough rest? What was the mean rest time between each assessment? What about between each trial?

Results

Overall you did a good job with the results. Here are my specific comments

1. Your tables were great! I especially liked Table 2. I believe instead of being in the supplementary section, both tables should be part of the main manuscript.

2. Can you please provide a CONSORT flow diagram of the participants involved in this study. How many were recruited? How many decided to participate? Also why were there 121 recorded, but only 112 used in this study? You can respond to that question in the methodology.

3. For the tables, should group 5 be <=?

Discussion

Overall this discussion did a good job of explaining how each measure was associated with the TUG however, I didn't quite get a takeaway from the discussion.

I believe you need to structure your discussion so that the reader has a takeaway message. I felt as if that was not very clear in the discussion.

Here are some more specific comments

1. In section 4.2, if you're going to talk about how knee torque was the most important factor you should focus on that and perhaps have another section dedicated to the other factors and their importance/lack there of.

2. I think somewhere in the discussion you should state that based on classification accuracy you should consider all 4 measures. However, you should perhaps talk about what's an acceptable percentage of accuracy and which combination/which one fall into it. As the discussion is structured right now it seems as if you are trying to state why they're important but you're not quite discussing the results in terms of which one/combination of which ones are the most important.

3. You should consider adding a paragraph on implications. What does all of this mean for someone who works in the geriatrics world?

6. PLOS authors have the option to publish the peer review history of their article (what does this mean?). If published, this will include your full peer review and any attached files.

Reviewer #1: No

Reviewer #2: No

Reviewer #3: **Yes: **Ali Boolani

---

## [Author Response · Author response to Decision Letter 0]

11 Jan 2021

PLOS ONE Academic Editor

Dear Manabu Sakakibara, Ph.D.

Thank you for giving us the opportunity to resubmit a revised version of our manuscript entitled “Using Machine Learning to Investigate the Relationship between Domains of Functioning and Functional Mobility in Older Adults”.

We have addressed all the comments of the reviewer to the best of our ability. We are grateful for the very detailed review of our previous manuscript and feel confident that addressing the reviewer’s comments has considerably improved the quality of this manuscript.

Reply to comments from Reviewer #1

Thank you very much for your careful review of our manuscript and helpful suggestions. We have revised the manuscript according to your comments; all your suggestions have been duly incorporated. We believe that the revised manuscript meets your desired level of improvement. In the following are our point-by-point responses to each of your comments.

〈Comment 1〉

1, The authors should improve the introduction of their study with clear aims and hypotheses. Additionally, comparing with previous studies and clarifying the reasons for method selection, what are the benefits and drawbacks of SVM? The explanation of SVM on page 4, lines 65-66 is incorrect. The kernel function is not the way to avoid overfitting.

Response: Thank you for important suggestions. We agree that aim, hypotheses and characteristics of SVMs were lacked in our introduction. We added and revised manuscript as follows.

Introduction

“To bridge the aforementioned gaps in the literature, we used support vector machines (SVMs) of machine learning classification algorithm (commonly used for estimating multivariate patterns) [24]. SVMs are suitable for finding relationships by high-dimensional mapping using support vectors from small sample data with complexity relationships.” (Lines 62–65 in page 4)

“The purpose of our study was clarified assessment patterns for each domain reflecting functional mobility rankings according to the TUG test. As such, we employed a machine learning classification by using TUG rank as an objective variable, while the assessment values of each domain (body structures, body functions, and activities) relative to the averages of the same generations were set as explanatory variables. From the viewpoint of structure of ICF, our hypothesis is that the prediction accuracy increases in the order of the domain from the bottom to the top, and the activities similar to TUG is the highest.” (Lines 69–75 in page 4)

2, The machine learning method SVM was used in this study. However, functional mobility prediction is a research area with plenty of effective methods and algorithms that can be applied to obtain a highly accurate classification. Therefore, whether other much transparent methods (for example, the methods Random Forest (RF), Artificial Neural Network (ANN) or Partial Least Square Discriminant Analysis (PLS-DA)) may achieve better prediction and interpretable results than the SVM method but not used in this study need to be carefully discussed.

Response: Thank you for your detailed suggestions. We completely agree with this advice that it is not just the kernel method of SVM that can avoid multicollinearity, but almost all machine learning prediction methods. Nonetheless, we have carefully rewritten our introduction and discussion so that our discussion of other potentially usable methods is not limited to SVMs. We have made the following changes considering your advice:

Introduction

“To bridge the aforementioned gaps in the literature, we used support vector machines (SVMs) of machine learning classification algorithm (commonly used for estimating multivariate patterns) [24]. SVMs are suitable for finding relationships by high-dimensional mapping using support vectors from small sample data with complexity relationships. Moreover, the prediction accuracy of machine learning prediction methods (e.g., Random Forest, Artificial Neural Network), including SVMs, are not affected by multicollinearity [25]. When dealing with multivariates that are correlated with each other such as in the present study, one of the reasonable methods to employ is SVMs [26]. ” (Lines 62–69 in page 4)

Discussion

“However, machine learning prediction can make it possible to analyze data with multicollinearity [35]. One of the methods of machine learning prediction is SVMs with kernel function. Therefore, results achieved through SVM projection are not affected by collinearity [36]. In other words, this machine learning prediction method enabled us to avoid removing important information about participants. Moreover, the input data are first projected onto a higher dimensional space before they are employed in the estimation process. Thus, this method allowed us to conduct a complex factor analysis by multiple variables.” (Lines 232–238 in page 12)

An SVM with high discrimination accuracy was suitable for this study due to its usefulness in selecting evaluations with complicated correlations. (Lines 240–241 in page 12)

Conclusion and Limitation

“In addition, there are various machine learning classification methods such as random forest and artificial neural network available today. It is necessary to identify and verify the optimal method for such assessments in future studies.” (Lines 292–294 in page 14)

References:

1) Kutner MH, Nachtsheim CJ, Neter J, Li W. Applied Linear Statistical Models (4th edition): McGrow-Hill / Irwin Series; 2004.

2) Cortes C, Vapnik V. Support-Vector Networks. Mach Learn. 1995;20:273-97.

3) Morlini I. On Multicollinearity and Concurvity in Some Nonlinear Multivariate Models. Statistical Methods and Applications. 2006;15(1):3-2

3, Moreover, a clear procedure of study should be claimed in the method part.

a) how many participants are exactly involved in as the input in the model? Because in the method section, the sample size was shown as 90, but in the results section, the input sample size was 121 or 112 (the statement was confusing);

Response: We apologize for this error and the confusion caused. We wanted to convey that we extracted 112 subjects from a database of 121 subjects. The required number of samples was calculated based on the previous study, and it was 90. We wanted to convey that the number of participants in this study cleared it. We have modified this discussion and included the selection criteria as shown below:

“The East Japan Community Study of Aging (EJCSA) is an ongoing longitudinal survey that targets home-dwelling, healthy older adults in eastern Japan. The number of data recorded in the database of the EJCSA from 2018 to 2020 was 121. A total of 112 subjects were extracted as the target data for analysis according to the criteria of aged over 50 years and without physical disabilities.” (Lines 82–86 in pages 4–5)

b) it was not clear how and why the sample size needs to be extended to 1000 and what the benefits are; since the present study only included a maximum of four variables.

Response: We apologize for our confusing expressions. Our discussion of the method used was missing a description of the method and purpose of resampling. We have now added the necessary details and rewritten the discussion as follows:

“The data of each group were randomly expanded to 1,000 data to overcome the limitation of a small sample size and difference in sample sizes among groups. Therefore, a total 5,000 data measurements from all groups were maintained for the distribution of actual data using the bootstrap method. This bootstrap resampling method is often used in demographic studies [33]. 

The data were randomly divided as 90% training data and 10% testing data. The training data were submitted to the SVM; that is, the SVM algorithm was constructed as a prediction model using the training data. After the training, the resultant SVM prediction model with the 90% training data (4500 data) was used to predictively classify the remaining 10% testing data (500 data) into either of the five TUG groups or the four assessments for cross-validation.” (Line 144–152 in page 7)

Reference: Li Y, Staley B, Henriksen C, Xu D, Lipori G, Winterstein AG. Development and validation of a dynamic inpatient risk prediction model for clinically significant hypokalemia using electronic health record data. Am J Health Syst Pharm. 2019;76(5):301-11

c) how the hyper-parameters of SVM were trained, which kernel function was used in the present study? Commonly, a valid hyper-parameter can be identified by using a validation set that is split from the 90% training set, or using a cross-validation method

Response: Thank you for this suggestion. As you said, our program was calculated based on a confusion matrix between actual and calculated data using a cross-validation method. Moreover, we used the Gaussian kernel function. We have added these details in the manuscript as shown below:

“This ensured that a trained SVM with a Gaussian kernel could prospectively be generalized (( ).” (Lines 157–158 in page 7)

d) whether all data as inputs to SVM model were standardized to make sure all variables have the same unit? Since multiple-unit variables can negatively impact the SVM model.

Response: We normalized the maximum value of each variable and then put it into the SVM model.

“All assessments were normalized by each maximum value and expressed as 0-1.” (Lines 135–136 in page 7)

4, The output parameters of the prediction models in Table 2 only show accuracy. However, in order to show the model is valid and not overfitting, a comprehensive evaluation with other validation matrices should be provided, such as sensitivity, specificity and AUC.

Response: Thank you for your kind advice. We agree that our statistical verification of the relationship between the number of parameters and the accuracy rate was insufficient. Thus, in accordance with your suggestion, we have added the statistical analysis with non-linear regression. We have added the details and changed the discussion as follows:

Methods

“The relationship between the accuracy rate and number of assessments was tested using AIC (Akaike's information criteria) as a non-linear regression equation.” (Lines 161–162 in page 8)

Results

“The relationship between accuracy rate and number of assessments was highly correlated (Supplemental Figure 2, R = 0.89, p < 0.001)” (Lines 172–173 in page 8)

Supplemental Figure 2

Figure 2. Relationships between Accuracy rate and Number of assessments

The circles show the accuracy rate of each combination of assessments and the dashed line represents the non-linear regression line.

Discussion

“The accuracy rate increased depending on the number of explanatory variables as assessments (Supplemental Figure 2).” (Lines 198–199 in page 11)

5, On page 8, line 173, the statement is incorrect, SVM is not a kernel function, it was a classification or prediction method includes a kernel function. How the SVM addresses the multicollinearity of data in the present study (e.g., linear separately in high dimensional space) should be interpreted.

Response: We agree that it is not just the kernel method of SVM that can avoid multicollinearity, but almost all machine learning prediction methods. Thus, we have carefully rewritten this part of the discussion to indicate this fact.

“However, machine learning prediction can make it possible to analyze data with multicollinearity [35]. One of the methods of machine learning prediction is SVMs with kernel function. Therefore, results achieved through SVM projection are not affected by collinearity [36].

” (Lines 223–225 in page 12)

6, On page 8, lines 176-177, the expression “SVM projection enabled us to use information that was removed when selecting variables” is unclear. Who removes which variables? Or you mean no need for a prior variables’ selection?

Response: We apologize for this confusing statement. We wanted to convey that when using multivariate analysis you have to remove the other explanatory variables, while when using machine learning prediction you do not have to remove the other explanatory variables. The necessary corrections to the text are shown below.

“In other words, this machine learning prediction method enabled us to avoid removing important information about participants. Moreover, the input data are first projected onto a higher dimensional space before they are employed in the estimation process. Thus, this method allowed us to conduct a complex factor analysis by multiple variables. Other studies have used SVMs to analyze the relationships between outcomes and multiple complicating factors, as to individually predict each participant’s prognosis.” (Lines 225–231 in page 12)

7, On page 9, line 190, the statement “a larger number of explanatory variables” is not proper because you only show four variables in the present study.

Response: We apologize for this inconsistency. We wanted to convey that the number of variables was dependent on the accuracy rate. For example, the accuracy rate was high in 4 variables but was low in 1 variable. We have modified the text as shown below:

“The accuracy rate increased depending on the number of explanatory variables as assessments (Supplemental Figure 2).” (Lines 198–199 in page 11)

8, On page 3, line 39, repeated references; lines 49-50, e.x. should be, e.g.,

Response: Thank you for pointing this out. We have modified the said description following your comment (Lines 39 and 49–50 in page 3).

Reply to comments from Reviewer #2

Thank you very much for your careful review of our manuscript and helpful suggestions. We have revised the manuscript according to your comments and have duly incorporated all your suggestions. We believe that the revised manuscript meets your desired level of improvement. The responses to each of your comments are as follows:

1. Line 77. Please provide information how these 90 subjects were sampled from the original study?

Response: Thank you for this comment and we apologize for the lack of clarity in this matter. We wanted to convey that we extracted 112 subjects from a database of 121 subjects. The required number of samples was calculated based on the previous study, and it was 90. We wanted to convey that the number of participants in this study cleared it. We have modified this discussion in the manuscript and included the selection criteria as shown below:

“The East Japan Community Study of Aging (EJCSA) is an ongoing longitudinal survey that targets home-dwelling, healthy older adults in eastern Japan. The number of data recorded in the database of the EJCSA from 2018 to 2020 was 121. A total of 112 subjects were extracted as the target data for analysis according to the criteria of aged over 50 years and without physical disabilities.” (Lines 82–86 in pages 4–5)

“We confirmed that the sample size was satisfied after being estimated using G*Power 3.1 software (Franz Faul, University of Kiel, Kiel, Germany), with an effect size 0.30, a minimum power 0.80, and α = 0.05.” (Lines 91–93 in page 5)

2. Line 137. 10% were used as a testing data. Considering small sample size, isn’t 10% of sample is too small for testing?

Response: Thanks for this question. We do not think that 10% of the total 5000 resampled data, which amounts to 500 testing data, is too small. However, we acknowledge that our expression of this was confusing, so we have added more detail and rewritten it as follows:

“The data of each group were randomly expanded to 1,000 data to overcome the limitation of a small sample size and difference in sample sizes among groups. Therefore, a total of 5,000 data measurements from all groups were maintained for the distribution of actual data using the bootstrap method. This bootstrap resampling method is often used in demographic studies [33]. 

The data were randomly divided as 90% training data and 10% testing data. The training data were submitted to the SVM; that is, the SVM algorithm was constructed as a prediction model using the training data. After the training, the resultant SVM prediction model with the 90% training data (4500 data) was used to predictively classify the remaining 10% testing data (500 data) into either of the five TUG groups or the four assessments for cross-validation.” (Lines 144–152 in page 7)

Reference: Li Y, Staley B, Henriksen C, Xu D, Lipori G, Winterstein AG. Development and validation of a dynamic inpatient risk prediction model for clinically significant hypokalemia using electronic health record data. Am J Health Syst Pharm. 2019;76(5):301-11

3. Any evaluation on the results with the testing data?

Response: We randomly divided the resampled data based on the mean and deviation into 10% and 90%. The resampling results are also shown in the Table 1. Therefore, we think that any special evaluation on the result of testing data was not necessary.

4. Line 151. “A total of 121 participants were included” which is confusing when line 77 says including 90 healthy subjects. Please clarify this.

Response: We apologize for this error. We extracted 112 subjects from a database of 121 subjects. Ninety (90) was calculated as the sample size. We have modified this explanation as shown below:

“The East Japan Community Study of Aging (EJCSA) is an ongoing longitudinal survey that targets home-dwelling, healthy older adults in eastern Japan. The number of data recorded in the database of the EJCSA from 2018 to 2020 was 121. A total of 112 subjects were extracted as the target data for analysis according to the criteria of aged over 50 years and without physical disabilities.” (Line 82–86 in page 4–5)

Reply to comments from Reviewer #3

Thank you very much for your careful review of our manuscript and helpful suggestions. We have revised the manuscript according to your comments and have duly incorporated all your suggestions. We believe that the revised manuscript meets your desired level of improvement. In the following are our point-by-point responses to each of your comments:

Methodology

1. Was this retrospective data that you used a SVM model for? Or did you go recruit participants from this longitudinal study to collect the data for this specific study? 

Response: We extracted 112 subjects from a database of 121 subjects. Ninety (90) was calculated by the sample size. We have modified this explanation as shown below:

“The East Japan Community Study of Aging (EJCSA) is an ongoing longitudinal survey that targets home-dwelling, healthy older adults in eastern Japan. The number of data recorded in the database of the EJCSA from 2018 to 2020 was 121. A total of 112 subjects were extracted as the target data for analysis according to the criteria of aged over 50 years and without physical disabilities.” (Line 82-86 in page 4)

2. If these are participants who you recruited from the longitudinal study how did you recruit them? How did you contact them? What was the eligibility criteria to be in this particular study?

However, if these participants were part of the data collected for this longitudinal study, how did you select the participants?

Response: We based the study on healthy elderly people extracted from a local government database, the East Japan Community Study of Aging. We extracted only the first survey data of 112 subjects aged over 50 years and without physical disabilities.

3. When was this data collected?

Response: We recorded these data from 2018 to 2020.

4. What was the procedure? What was the order in which the participants completed the assessments?

Response: Thank you for this question. We have added the said description following your comment.

“The participants completed the assessments randomly.” (Line 134-135 in page 4)

5. You state that the participants had enough rest between trials, what is considered enough rest? What was the mean rest time between each assessment? What about between each trial?

Response: Thank you for this question. The resting periods we used between trials were based on a previous study. We have corrected the said description following your comment.

“The participants had enough rest time between each trial, which involved resting for at least about 3 minutes between each trial, taking into account the effects of muscle fatigue.” (Lines 132–134 in page 6)

Reference:

1) Gandevia SC. Spinal and Supraspinal Factors in Human Muscle Fatigue: PHYSIOLOGICAL REVIEWS Vol. 81, No. 4, October 2001

Results

1. Your tables were great! I especially liked Table 2. I believe instead of being in the supplementary section, both tables should be part of the main manuscript.

Response: Thanks for your recommendation. We have placed them within the main text.

2. Can you please provide a CONSORT flow diagram of the participants involved in this study. How many were recruited? How many decided to participate? Also why were there 121 recorded, but only 112 used in this study? You can respond to that question in the methodology.

Response: Thank you for these questions and we apologize for the lack of clarity in these matters. We have rewritten the discussion on our recruiting and extracting methods as shown below. And, if we make a CONSORT, it will be this simple flow diagram. Please point out again if we should include it in manuscript.

“The East Japan Community Study of Aging (EJCSA) is an ongoing longitudinal survey that targets home-dwelling, healthy older adults in eastern Japan. The number of data recorded in the database of the EJCSA from 2018 to 2020 was 121. A total of 112 subjects were extracted as the target data for analysis according to the criteria of aged over 50 years and without physical disabilities.” (Lines 82–86 in pages 4–5)

Flow diagram

3. For the tables, should group 5 be <=?

Response: Unfortunately, we do not understand this question. Nonetheless, we offer the following explanation in hope that it answers your question. The subjects of Group 5 were able to complete the TUG assessment quickly, specifically, in less than 4.9 and 5.7 seconds for males and females, respectively.

Discussion

1. In section 4.2, if you're going to talk about how knee torque was the most important factor you should focus on that and perhaps have another section dedicated to the other factors and their importance/lack there of.

2. I think somewhere in the discussion you should state that based on classification accuracy you should consider all 4 measures. However, you should perhaps talk about what's an acceptable percentage of accuracy and which combination/which one fall into it. As the discussion is structured right now it seems as if you are trying to state why they're important but you're not quite discussing the results in terms of which one/combination of which ones are the most important.

Response: Thanks for your advice. We agree that there was a lack in discussing other factors beyond knee torque. To address this, in sub-section 4.2 we include a comparison between knee extension torque and other evaluations. We have listed the interpretations of the other assessments according to your recommendation in a summary of the results at the beginning of the discussion. However, as we mentioned in manuscript, we thought that we should not mention the causality between each assessment and functional mobility in this study. This is because the classification function obtained through the SVM is a black box that outputs only the classification results, making it difficult to interpret the contributions of each variables. We have added more details pertaining to this and modified the text as follows:

Discussion

“The accuracy rate increased depending on the number of explanatory variables as assessments (Supplemental Figure 2). This result is natural because all assessments are known to contribute to the TUG. Next, in combinations of three assessments, the OLST, FEV1 / FVC and SLM assessments were omitted in order. The accuracy rate of the combination of SLM, FEV1 / FVC, and OLST were lower than that of the combination of two assessments including KET. However, this accuracy rate did not depend on the number of assessments. Moreover, the bottom three in the single assessments were SLM, FEV1 / FVC, and OLST. From these points, the difference in the explanatory variables can be seen between SLM, FEV1 / FVC, and OLST compared to KET. However, since the chance of predicting the TUG’s five ranks by simply thinking is 20%, it cannot be said that the predictive accuracy of these evaluations is low (SLM: 56.6%; FEV1 / FVC: 50.8%; KET: 46.2%). This may be due to the fact that the evaluation is related with TUG.” (Lines 198–208 in page 11)

Conclusions and Limitations

“This study had some limitations. As previously mentioned, we could not include the causality between each domain/assessment and functional mobility through this study’s machine learning method. (Lines 278–280 in page 14)

3. You should consider adding a paragraph on implications. What does all of this mean for someone who works in the geriatrics world?

Response: Thanks for this suggestion. We added the specific implications (for physiotherapists) to the Conclusions and Limitations section as follows.

Conclusions and Limitations

“If physiotherapists investigate the possibility of falling among highly active elderly people, prioritizing the monitoring of items related to physical function, especially muscle weakness could be valid. In making a rehabilitation program to reduce fall risk among the elderly, increasing the proportion of strengthening or maintaining muscle strength could be valid as part of rehabilitation therapy. (Lines 290–294 in page 14)

---

## [Decision Letter · Decision Letter 1]

19 Jan 2021

Using machine learning to investigate the relationship between domains of functioning and functional mobility in older adults

PONE-D-20-33162R1

Dear Dr. Hirata,

We’re pleased to inform you that your manuscript has been judged scientifically suitable for publication and will be formally accepted for publication once it meets all outstanding technical requirements.

Kind regards,

Manabu Sakakibara, Ph.D.

Academic Editor

PLOS ONE

Additional Editor Comments (optional):

Reviewers' comments:

Reviewer's Responses to Questions

**Comments to the Author**

1. If the authors have adequately addressed your comments raised in a previous round of review and you feel that this manuscript is now acceptable for publication, you may indicate that here to bypass the “Comments to the Author” section, enter your conflict of interest statement in the “Confidential to Editor” section, and submit your "Accept" recommendation.

Reviewer #2: All comments have been addressed

Reviewer #3: All comments have been addressed

2. Is the manuscript technically sound, and do the data support the conclusions?

Reviewer #2: (No Response)

Reviewer #3: Yes

3. Has the statistical analysis been performed appropriately and rigorously? 

Reviewer #2: (No Response)

Reviewer #3: I Don't Know

4. Have the authors made all data underlying the findings in their manuscript fully available?

Reviewer #2: (No Response)

Reviewer #3: Yes

5. Is the manuscript presented in an intelligible fashion and written in standard English?

Reviewer #2: (No Response)

Reviewer #3: Yes

6. Review Comments to the Author

Reviewer #2: (No Response)

Reviewer #3: The authors have done a great job of responding to all of my comments and suggestions. I appreciate the thought that went into this revision.

7. PLOS authors have the option to publish the peer review history of their article (what does this mean?). If published, this will include your full peer review and any attached files.

Reviewer #2: No

Reviewer #3: **Yes: **Ali Boolani

---

## [Editor Report · Acceptance letter]

27 Jan 2021

PONE-D-20-33162R1 

Using Machine Learning to Investigate the Relationship between Domains of Functioning and Functional Mobility in Older Adults 

Dear Dr. Hirata:

I'm pleased to inform you that your manuscript has been deemed suitable for publication in PLOS ONE. Congratulations! Your manuscript is now with our production department. 

Kind regards, 

on behalf of

Dr. Manabu Sakakibara 

Academic Editor

PLOS ONE